# Effectiveness of Physical Exercise on Pain, Disability, Job Stress, and Quality of Life in Office Workers with Chronic Non-Specific Neck Pain: A Randomized Controlled Trial

**DOI:** 10.3390/healthcare11162286

**Published:** 2023-08-14

**Authors:** Yousef M. Alshehre, Shahul Hameed Pakkir Mohamed, Gopal Nambi, Sattam M. Almutairi, Ahmed A. Alharazi

**Affiliations:** 1Department of Physical Therapy, Faculty of Applied Medical Sciences, University of Tabuk, Tabuk 47512, Saudi Arabiaaalharazi@ut.edu.sa (A.A.A.); 2Department of Health and Rehabilitation Sciences, College of Applied Medical Sciences, Prince Sattam bin Abdulaziz University, Alkharj 11942, Saudi Arabia; 3Department of Physical Therapy, College of Medical Rehabilitation Science, Qassim University, Buraydah 52571, Saudi Arabia

**Keywords:** body awareness therapy, ergonomic, exercise therapy, neck pain, office workers

## Abstract

Neck pain is a widespread medical condition among office workers worldwide. This study aimed to compare physical exercises, including basic body awareness, neck-specific training exercises and ergonomic modifications, and ergonomic modifications alone in the management of chronic non-specific neck pain (NSNP) among office workers. Sixty participants were randomly allocated to an experimental group (physical exercises and ergonomic modifications) or a control group (ergonomic modifications) and received the intervention two times a week for eight weeks. The Numerical Pain Rating Scale, Neck Disability Index, Health and Safety Stress Tool, and Short Form Health Survey-36 were used to measure pain, disability, job stress, and quality of life at baseline, and at weeks 4 and 8 of the study period. A repeated measure ANOVA was used to determine the within-group significant differences and an independent *t*-test was utilized to compare group differences. The baseline data of the experimental and control groups showed no significant difference (*p* ≥ 0.05). The 2 × 3 mixed models ANOVA showed a significant difference in pain intensity (*p* = 0.001, *ηp*^2^ = 0.042), functional disability (*p* = 0.001, *ηp*^2^ = 0.052), work-related stress (*p* = 0.001, *ηp*^2^ = 0.036), and QoL (*p* = 0.012, *ηp*^2^ = 0.025). Four weeks post-intervention, the experimental group showed significant changes in primary (pain intensity and disability) (1.9; 95% confidence interval 1.65–2.14) and secondary (quality of life and work-related stress) outcomes (*p* < 0.001). The same gradual improvement in these variables was observed in the 8-week follow-up (*p* < 0.001). There was a significant improvement in clinical outcomes following the application of physical exercises with ergonomic modifications for chronic NSNP among office workers. This is significant for office workers because it suggests the importance of incorporating physical exercises into their daily routine and making ergonomic changes to their workspaces.

## 1. Introduction

Non-specific neck pain (NSNP) is a global health illness among office workers [1]. In 2020, neck pain impacted 203 million individuals worldwide. The global age-standardized prevalence of neck pain and years lived with disability was estimated to be 2450 per 100,000 individuals, which remained unchanged from 1990 to 2020 [2,3]. In addition, the burden of neck pain was 2.9% higher in females than males, with a peak prevalence between 45 and 74 years of age for both genders [4,5,6]. In office workers, the yearly prevalence of neck pain ranges from 42% to 63% [7]. In Saudi Arabia, the annual prevalence of neck pain was estimated to be 64% [5]. Neck pain and tenderness were the most common causes of absenteeism among office workers, which impairs their ability to function [4,8]. Computer professionals have 2 to 3 times more chances of having long-term neck pain than the general population [9]. An association between intense job strain and the onset of neck pain in office workers was reported [10]. Physical (or biological) exposure, such as prolonged sedentary and active muscular contraction, cervical pressure, extreme working conditions, a poor working environment, and tedious tasks, were reported to increase the possibility of neck pain [11]. Additionally, psychosocial factors, such as work-related stress and job strain, a lack of co-worker support, and decreased job security, were considered the causes of the rise in the incidence of neck pain [12,13]. Furthermore, neck pain has been linked to a decrease in health-related quality of life [14]. In 2007, the quality of life (QoL) scores of office employees in the Netherlands decreased by 31% after they started having neck pain [15].

Several physio-therapeutic techniques were used in treating neck pain, among them basic body awareness therapy (BBAT), a technique used by physical therapists to enhance patients’ postural awareness and control [16]. BBAT provides a wide variety of training exercises, including simple coordination of movement, and enhances the interaction of the body and mind by gradually increasing postural stability, flow, and awareness of body reactions and resources [17]. The session involves the patient and the therapist working together to complete exercises that are individually applied and performed in a variety of starting postures, such as lying, sitting, and standing [17].

Office workers who spend long hours in uncomfortable positions are more susceptible to develop musculoskeletal disorders, which can cause neck pain [18]. Ergonomic recommendations are suggested to prevent such difficulties, and musculoskeletal pain can be reduced with ergonomic modifications [19]. Mehrparvar et al. [20] investigated the positive effects of ergonomic interventions on musculoskeletal pain in terms of distress and work absenteeism [20]. Additionally, it has been demonstrated that ergonomic workplace interventions can reduce the cost of musculoskeletal problems while improving worker productivity and work efficiency [21]. A further option is to adapt the workplace layout in accordance with each employee’s anthropometrics in order to improve comfort and body posture and prevent musculoskeletal pain. To educate workers on the risks and necessary safety measures, these interventions may also include an educational component [21]. Frequent rest and physical activities during work are examples of ergonomic office interventions [21,22]. The severity, intensity, and frequency of musculoskeletal pain were found to be less frequent, and this was confirmed by a thorough evaluation of the effectiveness of chair modifications in relation to furniture changes [23].

Various therapy options have been established to effectively manage patients with job-related neck pain and expedite their return to work [24]. Evidence revealed that physical therapy, including exercise, mobilization, and electrotherapy modalities, reduces job-related neck pain and improves functional activities [25]. This is accomplished by retraining, strengthening, and stretching the muscles, mobilizing soft tissue, and enhancing workplace ergonomics and kinetic management. Resistance and endurance training reduced pain and disability values, for in-office employees, with chronic neck pain [26]. Office employees who underwent exercise had a substantial reduction in the intensity and duration of neck pain compared to those who got an educational brochure on workplace ergonomics [27].

Exercise therapy is one of the most commonly used treatments for NSNP in clinical practice; however, there is little evidence to support its efficacy in non-specific recurring patient populations, particularly those with neck pain. Gross et al. [28] performed a systematic review and indicated that there was no high-quality evidence to support the effectiveness of exercise for chronic neck pain among adults. In addition, the 2017 systematic review and meta-analysis by Chen et al. [7] recommend that future randomized controlled trials of ergonomic interventions should target office workers who are symptomatic and also stated that more research on neck pain prevention is essential. In different occupations, NSNP probably has different causes since individuals are exposed to different risk factors. Therefore, it would not be rational to prescribe the same exercises to everyone with neck pain. Most office workers are required to work long hours on a computer, putting more pressure on themselves. Insufficient empirical data exists to support the effectiveness of workplace-based interventions in preventing and reducing neck pain among office workers, and research on primary neck pain prevention is limited. Therefore, this randomized clinical trial aimed to investigate whether a physical exercise therapy program that focuses on a combination of therapeutic interventions, such as BBAT, neck-specific training exercises (strength, endurance, and stretching exercises), and ergonomic modifications would have greater improvement in reducing pain, disability, and job stress and improving QoL in office workers with chronic NSNP than those who received ergonomic modifications alone. In line with this, we hypothesized that there would be a significant difference between the two groups, such as that the physical exercise and ergonomic modification group would have greater improvement in pain intensity, disability level, job stress, and QoL in office workers with chronic NSNP than those who received ergonomic modifications alone.

## 2. Materials and Methods

### 2.1. Trial Design

This trial was a single-blinded, randomized, controlled trial that followed the CONSORT guidelines for reporting the data. The Declaration of Helsinki’s ethical principles were followed throughout the trial’s execution. The trial obtained ethical approval from the University of Tabuk (UT-153-31-2021) at Tabuk, Saudi Arabia, and is registered with ClinicalTrial.gov (NCT05725356). After signing a written informed consent form and prior to the baseline evaluation, sixty eligible participants were randomly allocated into one of two groups by using a computer-generated random sequence table, with 30 participants in the experimental group receiving physical exercise (BBAT and neck-specific training exercises) and ergonomic modifications and 30 participants in the control group receiving ergonomic modifications only. A physical therapist used an on-site computer system that hid the allocation to distribute the participants. The participants were informed that they would receive one of the two interventions, and they were not told which group they were assigned to. Only the physical therapists who administered the intervention knew which group each participant was in. It was difficult to blind the therapists who cared for the participants in this study due to the nature of the interventions. However, both groups received the ergonomic modifications for two sessions per week for eight weeks by an expert in occupational health who was unaware of which locations were assigned to the physical exercise and ergonomic modification group and which locations were assigned to the ergonomic modification group alone. An assigned physical therapist collected data on the primary and secondary outcome measures at baseline, four weeks, and eight weeks following the intervention and was blinded to each participant’s group allocation. The frequency and duration of the intervention were chosen to reflect typical physical therapy practice.

### 2.2. Participants

Participants (*n* = 60) were recruited between January and October 2022 from two governmental hospitals in Tabuk and Alkharj, Saudi Arabia. Potential participants were office workers with chronic NSNP who used computers. The inclusion criteria were as follows: any sex; age 25–60 years; office workers and computer users; the ability to continue working; and greater than three months of chronic NSNP. Exclusion criteria included a history of severe injury, previous physical therapy treatments, previous surgery, joint instability, frequent migraines, spasmodic torticollis, inflammatory rheumatic diseases, peripheral nerve entrapment, severe psychiatric illness, pregnancy, and other conditions that prevent physical loading.

### 2.3. Sample Size

The study predicted a 1-point difference in pain intensity between the groups, measured by the Numerical Pain Rating Scale (NPRS), with a standard deviation of 1.84 points. The specified parameters were 80% power and 5% alpha, with subsequent maximum dropouts of 15%. Consequently, 60 participants were selected for this study (30 in each group). The estimates used to determine the sample size were lower than the minimal clinically significant differences recommended, thereby improving the accuracy of the intervention efficacy estimates used in the computations [29].

### 2.4. Outcome Measures

The NPRS and Neck Disability Index (NDI), which measure neck pain intensity and disability, respectively, served as the study’s primary outcome measures, while the Health and Safety Executive (HSE)–Management Standards (Ms) Indicator Tool (HSE–MS instrument) and the Short Form Health Survey (SF-36), which measure job stress and quality of life, served as the study’s secondary outcomes. These measures are reliable and have established reference data.

#### 2.4.1. Numerical Pain Rating Scale (NPRS)

The NPRS is a valid and reliable tool used to assess pain intensity [30]. It uses the numbers from 0 to 10, and 0 indicates (no pain) through 1–3 (mild pain, slight impairment in daily life activities), 4–6 (moderate pain, substantial impairment in daily life activities), and 7–10 (severe pain, severe impairment in daily life activities) [30]. The participants were asked to rate their neck pain intensity level for the past seven days using the NPRS.

#### 2.4.2. Neck Disability Index (NDI)

The NDI is the most extensively used tool to measure the patient’s self-reported impairment due to neck pain [31]. It is a self-reported tool comprising 10 items: pain severity, personal care, lifting, work, headaches, concentration, sleeping, driving, reading, and recreation. Each answer is graded on a 6-point scale ranging from 0 (no disability) to 5 (severe disability). Some evaluators multiply the raw score by 2 and report the NDI on a 0–100% scale [31]. Higher scores indicate a greater degree of disability. MacDermid et al. [31] showed that the NDI is reliable and valid as an outcome measure for patients with neck pain. Participants were asked to fill out the NDI to rate their perceived functional disability.

#### 2.4.3. Health and Safety Executive (Hse)–Management Standards (Ms) Indicator Tool

The HSE–MS instrument is one of the most valid questionnaires for assessing psychosocial stress in the workplace [32]. It was developed by the UK Health and Safety Executive. Thirty-five questions are used to measure seven dimensions in the HSE–MS instrument. The seven dimensions are demands (including workload, work schedules, and work environment), control, supervisory support (including support and resources from the employer), peer support (including support and resources from colleagues), relationships (including effective measures to prevent conflict and deal with inappropriate behaviors), role (including a clear understanding of the employee’s role in the organization), and changes. A score of 1–2 suggests that there may be a problem and that the individual may be under stress as a result of demand-resource imbalance, while a score of 4–5 indicates that the individual has minimal problems with that management standard and so likely exhibits a low risk of imbalance. Every management standard receives the same score. The HSE is a valid and reliable tool for studying job stress [32]. All participants were asked to complete the HSE–MS instrument.

#### 2.4.4. The Short Form Health Survey (SF-36)

The SF-36 is a questionnaire used to measure QoL. Physical functioning (PF), general health (GH), bodily pain (BP), role physical (RP), social functioning (SF), role emotional (RE), vitality (VT), and mental health (MH) are the eight domains used to measure health over the past four weeks. The SF-36 has strong construct validity and can be used to assess patients with chronic NSNP [33]. SF-36 data were recorded at the baseline, fourth week, and eighth week.

### 2.5. Interventions

#### 2.5.1. Basic Body Awareness Therapy (BBAT)

In this study, BBAT was performed twice a week for eight weeks for the experimental group. The participants participated in 60-min BBAT group sessions during the following sessions. Every session included regular BBAT exercise therapy. Participants were instructed to concentrate on maintaining good posture, balance, uninterrupted breathing, improved awareness, and decreased unnecessary muscle tension while performing each activity. Using a set of exercises selected during the trial’s planning phase, the content of the BBAT was standardized. This strategy was recorded in writing for subsequent use. The treating physical therapist assisted the participants in modifying an exercise if they had difficulty with it. Breaks, restarts, multiple variations of the same exercise, and switching to other exercises from the written plan were used to achieve this. The physical therapist offered feedback and manual assistance but did so sparingly to encourage long-term recall of motor learning [34,35]. Any modifications to these exercises that the treating physical therapist chose to utilize for participants were their decision and were not predefined. The treating physical therapist was not allowed to add any new exercises that were not in the written plan.

#### 2.5.2. Neck-Specific Training Exercises

Physical therapists with five years of experience delivered neck-specific training in the hospital. Participants participated in two individual sessions each week for eight weeks during the intervention period. The exercise training program concentrated on improving neck muscular endurance, sensorimotor function, and pain management. Participants began with modest isometric neck movements to encourage the stimulation of the deep cervical muscles. The exercise program was then gradually advanced to low-load endurance exercises in the department. The purpose of the exercise was to make the muscles that support the neck and scapula stronger and more resilient. Even though the neck-specific training was the same for everyone, it was tailored to each participant based on how they responded to the exercises they chose and how quickly they improved. The physical therapists customized the neck-specific exercise program for each participant such that the exercises and the dosage corresponded with the participants’ present capacities. The participant’s training instructions included drawings illustrating the exercises, and they were instructed to inform their pain perception before and after each program session. The intensity of the exercise increased to the next level when the participant could perform the exercise with proper form and for the required number of repetitions and hold times. Training progressions ranged from solitary, low-load workouts to synergy exercises and endurance-strength exercises. The increment of the training depended on the participant’s pain status and neck movement quality and the completion of a specified number of sets and repetitions before progressing. The protocol for the neck-specific training exercises was two times a week for eight weeks [36,37]. No exercises were given to the control group participants.

##### Stretching Exercises

Neck and shoulder stretching exercises (shoulder rolling, trunk stretching, and back extension exercises) 10–15 repetitions, and a handout brochure on proper positions and daily ergonomics at work.

##### Strengthening Exercises

Exercises to strengthen the neck flexor muscles while sitting (front raises, lateral raises, horizontal flexion, extension, and shoulder extension with a TheraBand), 1 set of 15 repetitions); dynamic exercises (shrugs, presses, curls, bent-over rows, flies, and pullovers with dumbbells), 1 set of 20 repetitions; trunk and leg strengthening using body weight and multimodal rehabilitation; and recommended aerobic exercises, 3 sets of 20 repetitions.

##### Endurance Training

Lift the head up from a supine position to exercise the neck flexor muscles (3 sets of 20 reps); dynamic exercises (shrugs, presses, curls, bent-over rows, flies, dumbbell pullovers), 3 sets of 20 reps; body weight and multimodal rehabilitation to strengthen the trunk and legs, 3 sets of 20 reps; stretching the neck, shoulder, and upper limb muscles, 3 sets of 9 stretches; and advised aerobic exercises (as patient tolerance allows, up to 30 min).

#### 2.5.3. Ergonomic Modifications

Both groups were exposed to ergonomic modifications for eight weeks. The ergonomic modifications consisted of a “total workplace Occupational Safety and Health and ergonomic intervention” that included the alteration of the working desk and chair height, the sitting posture, and the distance and level between the eyes and the monitor following recommendations from the online rapid office strain assessment (ROSA) [38]. The workstations were modified in accordance with the ergonomic modifications. Individual anthropometric measurements were compared to furniture dimensions to find discrepancies. Using supports of varying heights, the monitor height was changed to match the demands of each worker. The monitor was right in front of the employee [38]. The suggested distance between the monitor and the eyes was 40 to 75 cm. The keyboard and mouse were arranged so that the forearm could rest on the table. The mouse was placed near the keyboard and aligned with the shoulder [38]. The ergonomic modifications were accomplished with the assistance of an expert in occupational health, who was unaware of which locations were assigned to the physical exercise and ergonomic modification group and which locations were assigned to the ergonomic modification group alone.

### 2.6. Statistical Analysis

The study participants’ demographic characteristics were documented, and the homogeneity of variance was calculated using Levene’s test. The data were depicted as the mean and standard deviation with a 95% confidence interval. The effects of treatment at various intervals were analyzed with a 2 × 3 mixed model analysis of variance (ANOVA) with an experimental and control group and specified time intervals (baseline, four weeks, and eight weeks). A repeated measure ANOVA was used to determine the significant difference within groups, and post hoc Tukey’s analysis was performed to find the difference between various intervals. An independent *t*-test was utilized to compare the groups. The statistical significance level was set at *p* < 0.05. For all statistical analyses, SPSS software (version 20.0; SPSS Inc., Chicago, IL, USA) was used.

## 3. Results

### Participants

Ninety-three participants were recruited for the study. The CONSORT flow chart in Figure 1 presents a comprehensive overview of the screening, enrollment, randomization, and analysis of the participants included in this study. Sixty participants with chronic NSNP were voluntarily selected as per the eligibility criteria and allocated equally into two groups. Following group allocation, one participant from each group did not complete the 8-week intervention program due to time constraints and personal reasons (Figure 1). In the study analysis, the intention-to-treat analysis was used.

The basic demographic variables, such as age, height, and weight, did not show a significant difference between the groups (*p* ≥ 0.05) at baseline. Males (60–67%) were affected by NSNP more than females in both groups. Causative factors, such as degeneration (37–40%), trauma (10–13%), inflammation (7–10%), strain (2%), and others (30–37%), led to chronic neck pain (Table 1).

Table 2 shows the mean and standard deviation (SD) of the variables for the two groups at three time intervals. The baseline data on pain intensity between the experimental and control groups did not show any statistical difference (*p* ≥ 0.05). The baseline data on functional disability, QoL, and overall work-related stress between the experimental and control groups did not show any statistical difference (*p* ≥ 0.05).

The 2 × 3 mixed model ANOVA showed a significant difference in pain intensity (*p* = 0.001, *ηp^2^* = 0.042), functional disability (*p* = 0.001, *ηp^2^* = 0.052), QoL (*p* = 0.012, *ηp^2^* = 0.025), and work-related stress (*p* = 0.001, *ηp^2^* = 0.036). Over four weeks of intervention, there was a significant improvement between the experimental (1.9; 95% confidence interval [CI] 1.65 to 2.14) and control groups (*p* = 0.001). A similar gradual improvement was seen in the 8-week follow-up period (2.4; 95% CI 2.28 to 2.51). The standard mean difference showed a greater percentage of improvement in pain intensity in the experimental group than in the control group.

The secondary variables, such as functional disability 14.01 (95% CI 9.24 to 18.77) and QoL −19.5 (95% CI −22.14 to −16.85) scores, showed a significant difference between the two groups after four weeks of intervention (*p* = 0.001). The same gradual improvement was seen in the 8-week follow-up in functional disability (11.6; 95% CI: 8.11 to 15.20) and QoL (−30; 95% CI: −33.19 to −26.80). Similarly, the work-related score between the experimental and control groups showed a significant difference (*p* = 0.001) during the 4-week and 8-week follow-up periods. Moreover, the percentage of improvement in each domain of work-related stress, such as demands (80.64%), control (58.55%), manager’s support (73.57%), peer support (73.27%), relationship (67.08%), role (82.89%), and change (70.41%), showed greater improvement in the experimental group than in the control group. The percentage of improvement was higher in the role component (82.89%) than in the control component (58.55%) of the work-related stress questionnaire (HSE–MS). Overall, the percentage of work-related stress was higher in the experimental group than in the control group. Table 2 provides further details on the comparison between groups.

Within groups, the repeated measures ANOVA followed by Tukey’s post hoc analysis showed a significant difference (*p* = 0.001) in improvement in pain intensity in both the experimental and control groups at four weeks and eight weeks. Furthermore, functional disability, QoL, and all the domains of work-related stress showed a significant difference (*p* < 0.001) at four weeks and eight weeks. Table 3 provides further details on the comparison within groups.

## 4. Discussion

The present study results revealed that the application of physical exercises, such as basic body awareness therapy (BBAT) and neck-specific training exercises with ergonomic modifications, showed better improvement than ergonomic modifications alone for neck pain among office workers over eight weeks. Physical exercise was shown to be an effective intervention for reducing pain intensity, functional disability, overall work-related stress, and improving QoL in office workers with chronic NSNP. In particular, combining physical exercise and ergonomic modifications was more effective than ergonomic modifications alone in improving the aforementioned clinical outcomes. The improvement in pain intensity, functional disability, QoL, and work-related stress was more pronounced in the experimental group than in the control group at four weeks and eight weeks of follow-up. These findings demonstrated that bodily awareness was defined as being in touch with sensations and emotions, having control over pain and other symptoms, and creating one’s identity through relationships with oneself and the patients. This demonstrates that the experimental group was familiar with the fundamental traits of BBAT. Their personal experiences using the BBAT method served as the foundation for body awareness. The relationship and the unique encounter, where each participant’s resources and opportunities were met and strengthened, were also mentioned by the participants as being significant. Patients with musculoskeletal disorders usually lack sensory-motor awareness, which manifests in faulty movement coordination, coping strategies, and activities of daily living [16]. There is no published evidence regarding the effectiveness of the physical exercise therapy program that focuses on a combination of therapeutic interventions, such as BBAT, neck-specific training exercises, and ergonomic modifications. However, this finding is in agreement with a review of previous research confirming that experiences of body awareness, including close touch with bodily emotions and sensations, may make it possible to reduce pain and other symptoms [15]. In our study, the patients’ reported levels of pain and disability gradually decreased in both groups at the fourth and eighth week of the intervention period.

An important component in determining the cause of neck pain in office employees is their psychosocial risk factor. BBAT, along with neck-specific training exercises, and ergonomic modifications, were shown to reduce stress in the experimental group in our study. The aim of BBAT relaxation training is to help individuals relax their muscles so that they may feel better and perform an activity with less effort [39]. The majority of the available research has mostly centered on exercise interventions, whereas comparatively little emphasis has been placed on ergonomic interventions [7]. The authors believe that exercise reduces stress levels and the harmful effects that follow from muscle tension and the release of stress-related chemicals into the bloodstream; it also promotes relaxation, increases blood and lymphatic circulation, enhances oxygen delivery to bodily tissues, and expedites the body’s clearance of toxins. Thus, it increases the body’s endorphin production, reduces fatigue, depression, anxiety, and improves sleep and QoL [40]. The present study revealed that neck-specific training exercises showed greater improvement in the experimental group due to the patients being exposed to a combination of treatments, such as stretching, strengthening, and endurance exercises. This result agrees with that of a systematic review of mechanical neck problems, which suggests that office employees with neck pain combine strengthening exercises with endurance or stretching exercises to ease pain [28].

A recent meta-analysis showed that physical exercise significantly improved general and mental health-related quality of life in office workers [41]. Additionally, a previous study revealed that healthy individuals who exercise experience an immediate analgesic effect [42]. With reduced pain and increased work and leisure activities, the patients’ life satisfaction and QoL improved [43]. The present study’s results show that applying physical exercises and ergonomic training improved both groups’ QoL, with greater improvement in the experimental group. A study by Stovner et al. [43] revealed that reducing and managing pain are very important ways to improve the QoL of individuals with neck pain. Pillastrini et al. [44] conducted a study on office employees with neck pain and stated that individualized and supervised ergonomic intervention with twice-a-month follow-up for five consecutive months, plus exercise training, reduced neck pain compared to a control group. Considering the above statement, our study’s results showed that basic body awareness and neck-specific training exercises, combined with ergonomic changes, were more effective at reducing neck pain, disability, and work-related stress factors and improving QoL over eight weeks of training. The strength of this study is that it is the first randomized, control study conducted in Saudi Arabia to determine the impact of physical exercises and ergonomic modifications in office workers with chronic NSNP on improving pain intensity, functional disability, quality of life, and work-related stress. In this study, first, the HSE–MS instrument was included as a limitation because, to the best of our knowledge, no validation research has been conducted in Saudi Arabia. Second, this study did not determine the long-term effects of these interventions, which could have provided more additional information about these interventions. The intervention period was only for eight weeks, and longer-term follow-ups are recommended. Third, the sample of participants was a convenience sample, which may restrict the generalizability of our findings. Lastly, associations between these variables were not found, which may provide more information about these interventions and NSNP. Therefore, the authors believe that more research on HSE–MS in the Saudi population, with long-term measurement of these interventions, is required to overcome these limitations.

## 5. Conclusions

The reports of this trial have clinical significance for neck pain prevention measures in the workplace. The study adds to the existing literature by providing evidence that combining the BBAT, neck-specific training exercises, and ergonomic training would be an effective intervention in lowering pain levels, reducing disability, improving QoL, and reducing work-related stress symptoms for office workers with chronic NSNP. By implementing these interventions, companies, healthcare professionals, and policymakers can help to reduce stress and improve the QoL of office workers. This can lead to increased productivity, decreased absenteeism, and improved employee morale.

## Figures and Tables

**Figure 1 healthcare-11-02286-f001:**
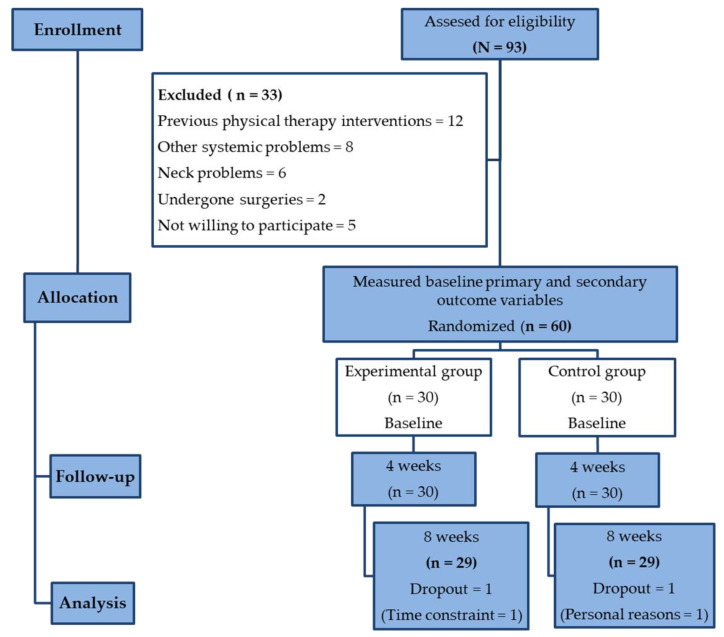
Consort flow chart of participants’ screening, enrollment, randomization, and analysis.

**Table 1 healthcare-11-02286-t001:** Mean ± standard deviation (95% confidence interval) and number of cases (percentages) of demographic details of the experimental and control groups.

Variable	Experimental Group (*n* = 30)	Control Group (*n* = 30)	Between-Group *p*-Value
Age (y)	42.1 ± 4.5	44.2 ± 4.8	0.085
Gender			
Male	18 (60%)	20 (67%)	0.592
Female	12 (40%)	10 (33%)	
Marital status			
Married	26 (87%)	25 (83%)	0.717
Single	4 (13%)	5 (17%)	
Height (cm)	171.2 ± 4.8	172.3 ± 4.9	0.383
Weight (kg)	68.8 ± 5.9	71.1 ± 6.2	0.146
Smoking			
Yes	16 (53%)	15 (50%)	0.796
No	14 (47%)	15 (50%)	
Dominant hand			
Right	21 (70%)	22 (73%)	0.774
Left	9 (30%)	8 (27%)	
Working hours per day			
6–8	21 (70%)	22 (73%)	0.894
9–10	6 (20%)	6 (20%)	
More than 10	3 (10%)	2 (7%)	
Working experience (y)			
1–5	2 (7%)	1 (3%)	0.826
6–10	3 (10%)	3 (10%)	
11–15	7 (23%)	5 (17%)	
More than 15	18 (60%)	21 (70%)	
Causative factors			
Inflammation	3 (10%)	2 (7%)	0.940
Degeneration	12 (40%)	11 (37%)	
Traumatic	4 (13%)	3 (10%)	
Strain	2 (7%)	3 (10%)	
Others	9 (30%)	11 (37%)	
Occurrence of pain			
One time	8 (27%)	6 (20%)	0.541
More than one time	22 (73%)	24 (80%)	
Pain during sleep			
Yes	18 (60%)	19 (63%)	0.790
No	12 (40%)	11 (37%)	

*p*-value unpaired Student’s *t*-test for quantitative variables and Chi-squared tests for qualitative variables.

**Table 2 healthcare-11-02286-t002:** Pre- and post-primary and secondary outcome measures analysis between the experimental and control groups.

Variable	Duration	Experimental Group	Control Group	Time × Group *p*-Value	*p*-Value
Pain intensity (NPRS)	Baseline	7.2 ± 0.6	7.3 ± 0.7	0.001 *	0.578
	4 weeks	3.8 ± 0.4	5.7 ± 0.5		0.001 *
	8 weeks	1.8 ± 0.1	4.2 ± 0.3		0.001 *
	% of improvement	75%	42.47%		
Functional disability(NDI)	Baseline	51.86 ± 10.7	52.11 ± 11.2	0.001 *	0.934
	4 weeks	32.12 ± 7.2	46.13 ± 10.1		0.001 *
	8 weeks	25.12 ± 5.2	36.78 ± 8.2		0.001 *
	% of improvement	51.57%	29.42%		
Quality of life (SF-36)	Baseline	38.8 ± 4.1	39.9 ± 4.2	0.001 *	0.308
	4 weeks	63.8 ± 5.5	44.3 ± 4.7		0.001 *
	8 weeks	82.5 ± 7.1	52.5 ± 5.1		0.001 *
	% of improvement	71.4%	20.96%		
Work-related stress (Demands)	Baseline	1.28 ± 0.12	1.31 ± 0.13	0.001 *	0.356
	4 weeks	2.82 ± 0.35	1.99 ± 0.29		0.001 *
	8 weeks	4.28 ± 0.41	2.23 ± 0.29		0.001 *
	% of improvement	80.64%	24.93%		
Work-related stress (Control)	Baseline	1.26 ± 0.25	1.24 ± 0.12	0.001 *	0.694
	4 weeks	2.34 ± 0.32	1.92 ± 0.18		0.001 *
	8 weeks	3.45 ± 0.39	2.21 ± 0.20		0.001 *
	% of improvement	58.55%	25.79%		
Work-related stress (Manager’s support)	Baseline	1.67 ± 0.16	1.63 ± 0.14	0.001 *	0.307
	4 weeks	3.12 ± 0.32	1.89 ± 0.17		0.001 *
	8 weeks	4.12 ± 0.43	2.21 ± 0.20		0.001 *
	% of improvement	73.57%	17.21%		
Work-related stress (Peer support)	Baseline	1.67 ± 0.16	1.63 ± 0.16	0.001 *	0.336
	4 weeks	2.45 ± 0.31	1.95 ± 0.19		0.001 *
	8 weeks	4.11 ± 0.39	2.21 ± 0.22		0.001 *
	% of improvement	73.27%	17.21%		
Work-related stress (Relationship)	Baseline	1.84 ± 0.18	1.83 ± 0.17	0.001 *	0.825
	4 weeks	2.34 ± 0.29	1.98 ± 0.19		0.001 *
	8 weeks	3.96 ± 0.38	2.25 ± 0.23		0.001 *
	% of improvement	67.08%	13.24%		
Work-related stress (Role)	Baseline	1.96 ± 0.20	1.93 ± 0.18	0.001 *	0.548
	4 weeks	3.82 ± 0.32	2.23 ± 0.24		0.001 *
	8 weeks	4.48 ± 0.40	2.43 ± 0.26		0.001 *
	% of improvement	82.89%	16.28%		
Work-related stress (Change)	Baseline	1.35 ± 0.14	1.33 ± 0.12	0.001 *	0.554
	4 weeks	2.63 ± 0.26	1.89 ± 0.27		0.001 *
	8 weeks	3.92 ± 0.38	2.20 ± 0.20		0.001 *
	% of improvement	70.41%	23.70%		

* Significant, NPRS—Numerical Pain Rating Scale, NDI—Neck Disability Index, HSE—Health and Safety Executive Standards, SF-36—Short form 36.

**Table 3 healthcare-11-02286-t003:** Pre- and post-primary and secondary outcome measures analysis within the experimental and control groups.

Variable/Time		Mean Difference CI 95% (Upper Limit–Lower Limit)		
		Baseline × 4 Weeks	Baseline × 8 Weeks	4 Weeks × 8 Weeks
Pain intensity (NPRS)	Experimental group	−3.4 (−3.65 to −3.14) *p* = 0.001	−5.4 (−5.65 to −5.14) *p* = 0.001	−2.0 (−2.25 to −1.74) *p* = 0.001
	Control group	−1.6 (−1.92 to −1.27) *p* = 0.001	−3.1 (−3.42 to −2.77) *p* = 0.002	−1.5 (−1.82 to −1.17) *p* = 0.001
Functional disability (NDI)	Experimental group	−19.7 (−24.6 to −14.7) *p* = 0.001	−26.7 (−31.6 to −21.7) *p* = 0.001	−7.0 (−11.9 to −2.05) *p* = 0.003
	Control group	−5.9 (−12.0 to 0.12) *p* = 0.056	−15.3 (−21.4 to −9.2) *p* = 0.001	−9.3 (−15.4 to −3.2) *p* = 0.001
Quality of life (SF-36)	Experimental group	25.0 (21.4 to 28.5) *p* = 0.001	43.7 (40.1 to 47.2) *p* = 0.001	18.7 (15.1 to 22.2) *p* = 0.001
	Control group	4.4 (1.51 to 7.28) *p* = 0.001	12.6 (9.71 to 15.4) *p* = 0.001	8.2 (5.31 to 11.0) *p* = 0.001
Work-related stress (Demands)	Experimental group	1.54 (1.34 to 1.73) *p* = 0.001	3.0 (2.80 to 3.19) *p* = 0.001	1.46 (1.26 to 1.65) *p* = 0.001
	Control group	0.68 (0.52 to 0.83) *p* = 0.001	0.92 (0.76 to 1.07) *p* = 0.001	0.24 (0.08 to 0.39) *p* = 0.001
Work-related stress (Control)	Experimental group	1.08 (0.87 to 1.28) *p* = 0.001	2.19 (1.98 to 2.39) *p* = 0.001	−2.0 (−2.25 to −1.74) *p* = 0.001
	Control group	0.68 (0.57 to 0.78) *p* = 0.001	0.97 (0.86 to 1.07) *p* = 0.001	0.29 (0.18 to 0.39) *p* = 0.001
Work-related stress (Manager’s support)	Experimental group	1.45 (1.25 to 1.64) *p* = 0.001	2.45 (2.25 to 2.64) *p* = 0.001	1.0 (0.80 to 1.19) *p* = 0.001
	Control group	0.26 (0.15 to 0.36) *p* = 0.001	0.58 (0.47 to 0.68) *p* = 0.001	0.32 (0.21 to 0.42) *p* = 0.001
Work-related stress (Peer support)	Experimental group	0.78 (0.59 to 0.96) *p* = 0.001	2.44 (2.25 to 2.62) *p* = 0.001	1.66 (1.47 to 1.84) *p* = 0.001
	Control group	0.32 (0.20 to 0.43) *p* = 0.001	0.58 (0.46 to 0.69) *p* = 0.001	0.26 (0.14 to 0.37) *p* = 0.001
Work-related stress (Relationship)	Experimental group	0.50 (0.31 to 0.68) *p* = 0.001	2.12 (1.93 to 2.30) *p* = 0.001	−2.0 (−2.25 to −1.74) *p* = 0.001
	Control group	0.15 (0.02 to 0.27) *p* = 0.011	0.42 (0.29 to 0.54) *p* = 0.001	0.27 (0.14 to 0.39) *p* = 0.001
Work-related stress (Role)	Experimental group	1.86 (1.66 to 2.05) *p* = 0.001	2.52 (2.32 to 2.71) *p* = 0.001	0.66 (0.46 to 0.85) *p* = 0.001
	Control group	0.30 (0.15 to 0.44) *p* = 0.001	0.50 (0.35 to 0.64) *p* = 0.001	0.20 (0.05 to 0.34) *p* = 0.003
Work-related stress (Change)	Experimental group	1.28 (1.10 to 1.45) *p* = 0.001	2.57 (2.39 to 2.74) *p* = 0.001	1.29 (1.11 to 1.46) *p* = 0.001
	Control group	0.56 (0.43 to 0.68) *p* = 0.001	0.87 (0.74 to 0.99) *p* = 0.001	0.31 (0.18 to 0.43) *p* = 0.001

## Data Availability

Data can be obtained by contacting the corresponding author.

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
