# Peer review of "Effectiveness of Physical Exercise on Pain, Disability, Job Stress, and Quality of Life in Office Workers with Chronic Non-Specific Neck Pain: A Randomized Controlled Trial"

_healthcare, 2023, doi:10.3390/healthcare11162286_

Round 1
Reviewer 1 Report (New Reviewer)
Dear authors,
I had the opportunity to review the paper entitled “Effectiveness of Physical Exercise on Job Stress and Quality of Life in Office Workers with Chronic Non-Specific Neck Pain: A Randomized Controlled Trial”
After reading the manuscript I have some concerns that should be resolved:
Abstract:
It does not mention the duration or intensity of the exercises, making it difficult for readers to understand the intervention fully.
While the abstract mentions that there is a significant improvement in clinical outcomes, it does not clearly state the specific implications or practical applications of these findings.
Introduction:
Repetitive Information: The introduction contains repetitive information, such as the high prevalence of non-specific neck pain (NSNP) among office workers, its definition, and the statistics regarding its occurrence. This information could be condensed and presented more concisely.
Lack of Clear Research Gap: While the introduction highlights the prevalence and impact of NSNP, it fails to clearly outline the research gap or the specific aim of the study. The introduction should provide a strong rationale for conducting the study and highlight the need for the current investigation.
What was your originality?
Method:
Lack of Randomization Details: The method should provide more specific information about the randomization process, such as the method used and whether it was concealed.
Blinding Issue: The method mentions single-blinding but does not explain how blinding was achieved or which parties were blinded during the study.
Missing Outcome Measure Descriptions: The method briefly mentions the outcome measures without providing a detailed description of each tool's reliability, validity, and relevance to the study.
Results and discussion:
Structure of Results: The current structure of presenting the results seems disorganized, with data and statistics scattered throughout the section. The data should be logically organized, and the statistical tests and outcomes should be clearly presented.
Lack of Context in Discussion: The "Discussion" section could benefit from providing more context and relating the findings to existing literature. A more comprehensive analysis and interpretation of the results are needed.
The "Discussion" section lacks specific recommendations for future research or practical applications based on the study's results. Adding practical implications will enhance the article's impact.
Some sentences and phrases contain errors in grammar and syntax, making the text confusing and challenging to read. Improving the sentence structure will enhance readability.
Author Response
Please see the attachment.

Reviewer 2 Report (New Reviewer)
Please see the attachment.

Author Response
Please see the attachment.

Reviewer 3 Report (New Reviewer)
This article investigates the role of a combination of Basic Body Awareness Therapy, specific neck exercises and ergonomic training in the treatment of neck pain in office workers. In doing so, they compare it with an intervention based on ergonomic training alone. The topic is interesting, but it is not new (see, for example, the 2017 systematic review and meta-analysis by Chen et al., https://doi.org/10.1093/ptj/pzx101) and, moreover, it does not differentiate the possible influence of each of the therapies used on neck pain.
On the positive side, the manuscript is clear, well-structured, has a good experimental design, complies with research ethics and has easy-to-interpret tables of results.
As areas for improvement, I would like to suggest the following:
- Pain seems to be the primary outcome, as stated in the abstract itself, but the title of the article highlights two of the secondary variables: quality of life and work-related stress. I think this should be changed.
- The Introduction is a bit long and 16/28 references are from 2013 or earlier. Perhaps a more up-to-date bibliography should be used. For example, on the Global Burden of Disease and neck pain (reference 6), Shin et al. from 2022 (doi: 10.3389/fneur.2022.955367) or Safiri et al. from 2020 (doi: 10.1136/bmj.m791) or Wu et al. from 2023 (http://dx.doi.org/10.2139/ssrn.4427441) could be used. For the prevalence of neck pain (reference 7), Kazeminasab et al., 2022 (doi https://doi.org/10.1186/s12891-021-04957-4), or Alhakami et al, 2023 (doi https://doi.org/10.3390/healthcare10071320) could be considered.
On the other hand, the sentence in lines 56 to 60 does not seem to be in the most appropriate place.
- In Material and Methods regarding the research design, I think it should be justified to choose 8 weeks of intervention. One might think that it is a bit short. In addition, it seems to me a limitation of the study that there is no follow-up to see whether and to what extent the improvements remained in the short or medium term.
In addition, it would be clearer to establish which are primary and which are secondary outcome measures (2.3 section).
- In relation to the Results, I think that the text repeats information that appears in the tables (for example, the first paragraph, between lines 270 and 283). I think it is better not to do so.
I also think that Figure 2 is not necessary, as its information already appears in Table 2.
- In relation to the Discussion, it seems to me to be somewhat scarce. It compares little with other studies, being a subject on which there are quite a few publications. Furthermore, I think the authors should revise some statements, such as the one on lines 351-352 (BBAT has been shown to reduce stress in our study in the experimental group), as the improvements seen in the study are not due to a single therapy, but to a combination of therapies.
In addition, I think the 6 point of the article (lines 384-395) should go within the section of the Discussion.
- Finally, in relation to the Bibliography, I think it should be updated (as I have already mentioned). In addition, references 29 and 30 are the same (one should be deleted and renumbered) and, when a study is cited in the text, the reference number should follow the author (for example, on line 75 it should be Mehrparvar et al. [21], and without the initials of the name, or on line 164 MacDermid et al. [31], and without the year of publication; also correct on lines 369 and 371).
Round 2
Reviewer 1 Report (New Reviewer)
The authors have responded to my comments. No further questions.
This manuscript is a resubmission of an earlier submission. The following is a list of the peer review reports and author responses from that submission.
Round 1
Reviewer 1 Report
The manuscript “Effectiveness of Physical Exercise on Job Stress and Quality of Life in Office Workers with Chronic Non-Specific Neck Pain: A Randomized Controlled Trial” aims to investigate whether exercise therapy that focuses on therapeutic interventions and ergonomic modifications can help office workers with chronic non-specific neck pain reduce pain, disability, and job stress and improve their quality of life.
In order to guide the authors, the following suggestions are proposed:
Throughout the text the authors speak interchangeably of neck pain and non-specific neck pain, I think this should be made clear. Further literature review is recommended to improve the introduction and discussion. Some of the information given in the introduction is not cohesive, it gives the impression of putting one idea next to another but without textual cohesion.
In the abstract it is not clear what the aim of the study is.
The text does not include hypotheses.
Lines 38-41: The authors repeat the word “condition” three times. Use of synonyms is recommended.
Lines 42-44: Why do the authors only give Saudi Arabia and Australia's prevalence of non-specific neck pain? Why Australia? What is happening elsewhere in the world?
Line 55: Authors should further develop this technique: basic body awareness therapy. And give information on BBAT and ergonomics in separate paragraphs as these are two different interventions. Has BBAT previously been used in neck pain?
Lines 58-71: It is recommended to order the information on ergonomics, the text lacks cohesion.
In the paragraph beginning on line 72, the authors talk about quality of life and then turn to therapy options. Authors should give coherence to the text.
Line 112: The blinding of patients is not clear. “The participants (blinded individuals) were informed that they would receive one of the two interventions but were unaware of which therapy they were receiving”. How can patients be blinded to an exercise intervention?
Line 117: “Both groups received the concerned intervention for two sessions per week, for eight weeks”. This sentence is not clear for me. Then I have read in the methods section, line 219 “Both groups were exposed to ergonomic training for eight weeks”, but I don't know what you mean by this because it is not explained.
The exclusion criteria are unclear. Previous surgery should be included as an exclusion criterion as the authors exclude two patients with surgery, and surgery should be removed from table 1.
On which previous study have you based the sample size calculation? Please, add the reference.
Lines 147-148: “Throughout the seven days of the study, the participants were asked to rate their pain intensity level”. What do the authors mean by the 7 days of the study?
Line 155: “Some evaluators multiply the raw score by 2 and report the NDI on a 0–100% scale”. Lack of reference in this statement.
Is the HSE-MS instrument validated for the Saudi Arabian population? If not, it should be explained as a limitation.
Line 168-170: “In this study, a higher score on each dimension (5 points) indicated a lower stress level (1 point)”. I do not understand this sentence. Authors should explain better this instrument.
The statistical analysis section is not complete, Tukey's post-hoc test is missing (this information appears in the abstract). It is also unclear how the authors calculate the % improvement in the variables that it is included in table 2.
I do not quite understand why the authors study homogeneity using the Kolmogorov-Smirnov test. Could you explain it to me? Have you performed normality tests? Have the assumptions of ANOVA performance been checked?
Table 1 does not provide p-values for qualitative variables; the authors should add them. Also, they should include 95% confidence interval for quantitative variables. It is recommended to include in the title of table 1: mean ± standard deviation and (95% confidence interval) or number of cases (percentages). And at the foot of the table, it should be included: p-value unpaired Student’s t-test for quantitative variables and Chi-squared tests for qualitative variables.
On the other hand, it is recommended to delete the first column of the table 1 as it does not contribute anything to the table. The presentation of the variable name column is not very accurate, it does not allow to distinguish the different categories of variables.
Table 1 shows variables that have not been explained in the methods section.
Table 2:
- - Could authors explain what are the time x group p-value and the p-value columns? I cannot understand it.
- - I recommend deleting column “Sr. No”.
- - Where are Tukey's post-hoc results?
The discussion does not focus on the results obtained. It is poorly elaborated. The authors do not discuss the limitations or strengths of their study.

Author Response
"Please see the attachment."

Reviewer 2 Report
Thank you for the opportunity to review this article. This article is focusing on a relevant topic, because neck pain affects the majority of the population, office workers are the most susceptible to this condition.
I appreciate the authors' work, the study design is very clearly described, as well as the different types of interventions. In my opinion, it only needs some minor changes to be published, which can improve its quality.
These are my comments for the authors:
- Sample size is certainly a limitation in this study, I think it needs to be highlighted by the authors in the text.
- The duration of the intervention is rather short (2/week for 8 weeks) but the results obtained are significant. Considering the need globally to establish effective and sustainable models of intervention in the treatment of chronic conditions, I recommend that the authors explore this aspect, which may be the real "news" of this study. The beneficial effects of exercise even on chronic patients are well known in the literature, while identifying sustainable intervention models is an urgent need.
- As the authors state in chapter 2.5.2 Neck-Specific Training Exercises, the role of the therapist to tailor the exercise program to the needs of each patient was crucial. I encourage the authors to emphasize this concept, because the training of specialized personnel in the treatment of chronic conditions is essential for the well-being of patients.
- I invite the authors to make the acronym CNP (chronic neck pain?) explicit in line 84.
- I suggest the authors edit Figure.2, which is currently very difficult to read.
Author Response
"Please see the attachment."

Reviewer 3 Report
Review Report
The authors submitted a manuscript aiming to investigate whether exercise therapy that focuses on therapeutic interventions, can improve health related outcomes in office workers.
The manuscript is of interest to the readership of the journal.
There are, however, some concerns that I will present in this review report.
Abstract
In lines 28-32, when presenting the results, the interaction time x Group is worth mentioning.
Has the protocol been registered in any registering platform? If not, why?
Introduction
The topic is well framed and the reader is easily led to understand the aims of the study.
Methodology
Regarding sample size calculation. Please provide adequate citation for the assumptions regarding the predicted pain differences. It is not clear if reference [29] is the one that should be cited and this must be clarified.
I felt the need to know more about the BBAT exercise therapy and the neck-specific training exercises. Please provide more information regarding exercises and progression plan.
Regarding that analysis, what assumptions were verified before the repeated measures ANOVA was used?
Discussion
Please report the limitations of the study.
Conclusion
Please provide suggestions for future studies.
Congratulations for your work.
Author Response
"Please see the attachment."

Round 2
Reviewer 1 Report
The authors have submitted the text with change control and line numbering errors that have made revision very difficult.
There are still problems of coherence of ideas, lack of cohesion within paragraphs.
They continue to talk about patients being blinded despite having received an intervention consisting of physical exercise. When the patient reads the informed consent, he/she knows that there are several groups in the study and the therapy is carried out if he/she is in the intervention group, so it seems that the authors do not understand what it means to say that they were blinded.
“In line with this, we hypothesized that there is no significant difference in job stress and quality of life between physical exercises and ergonomic modifications in office workers with chronic NSNP”. The hypothesis is not clear.
I don't understand why they have changed the sample size calculation section when I only asked them for the reference. In the previous version it was clearer as I don't understand that the result of the calculation tells them that they need 1 patient. It does not make sense to me.
The intervention protocol has no references.
On a statistical level they do not answer my questions, they tell me that there is no need for Tuckey's post hoc and yet they have 3 time points at which they assess patients. Besides I think there is confusion between normality and homocedasticity.
Not all the recommendations made, such as including all the variables given as results in the method, have been followed.
When the size of a sample is calculated from data collected from tests, and it is done correctly, the sample size cannot be said to be a limitation.
On the other hand, the way limitations are worded does not seem to me to be appropriate. It is also recommended to include the strengths that the authors believe their work has.
If there is no evidence to compare results in this pathology, they could have explored what happens with this therapy and other conditions.